# Inflammatory pathways amongst people living with HIV in Malawi differ according to socioeconomic status

**Christine Kelly**[1,2,3]*, **Willard Tinago**[1], **Dagmar Alber**[4], **Patricia Hunter**[4],
**Natasha Luckhurst**[5], **Jake Connolly**[5], **Francesca Arrigoni**[5], **Alejandro Garcia Abner**[1],
**Raphael Kamn'gona**[2], **Irene Sheha**[2], **Mishek Chammudzi**[2], **Kondwani Jambo**[2],
**Jane Mallewa**[6], **Alicja Rapala**[7], **Patrick W. G. Mallon**[1], **Henry Mwandumba**[2], **Nigel Klein**[4],
**Saye Khoo**[3]

**1** Cente for Experimental Pathogen Host Research (CEPHR), University College Dublin, Dublin, Ireland,
**2** Malawi Liverpool Wellcome Trust Clinical Research Program, Blantyre, Malawi, **3** Institute of Translational
Medicine, University of Liverpool, Liverpool, United Kingdom, **4** Institute of Child Health, University College
London, London, United Kingdom, **5** Kingston University, London, United Kingdom, **6** College of Medicine,
University of Malawi, Blantyre, Malawi, **7** Institute of Cardiovascular Science, University College London,
London, United Kingdom

* christine.kelly@ucd.ie

**Data Availability Statement:** All relevant data are within the manuscript and its Supporting information files.

## Abstract

### Background

Non-communicable diseases (NCDs) are increased amongst people living with HIV (PLWH) and are driven by persistent immune activation. The role of socioeconomic status (SES) in immune activation amongst PLWH is unknown, especially in low-income sub-Saharan Africa (SSA), where such impacts may be particularly severe.

### Methods

We recruited Malawian adults with CD4<100 cells/ul two weeks after starting ART in the REALITY trial (NCT01825031), as well as volunteers without HIV infection. Clinical assessment, socioeconomic evaluation, blood draw for immune activation markers and carotid femoral pulse wave velocity (cfPWV) were carried out at 2- and 42-weeks post-ART initiation. Socioeconomic risk factors for immune activation and arterial stiffness were assessed using linear regression models.

### Results

Of 279 PLWH, the median (IQR) age was 36 (31–43) years and 122 (44%) were female. Activated CD8 T-cells increased from 70% amongst those with no education to 88% amongst those with a tertiary education (p = 0.002); and from 71% amongst those earning less than 10 USD/month to 87% amongst those earning between 100–150 USD/month (p = 0.0001). Arterial stiffness was also associated with higher SES (car ownership p = 0.003, television ownership p = 0.012 and electricity access p = 0.029). Conversely, intermediate monocytes were higher amongst those with no education compared to a tertiary education

**Funding:** The source of funding for this study was Wellcome Trust Training Fellowship grant number 099934/Z/12/A. The funders had no role in study design, data collection and analysis, decision to publish, or preparation of the manuscript. The grant holder, Christine Kelly, received a salary from this fellowship.

**Competing interests:** The authors have declared that no competing interests exist.

(12.6% versus 7.3%; p = 0.01) and trended towards being higher amongst those earning less than 10 USD/month compared to 100–150 USD/month (10.5% versus 8.0%; p = 0.08). Water kiosk use showed a protective association against T cell activation (p = 0.007), as well as endothelial damage (MIP1β, sICAM1 and sVCAM1 p = 0.047, 0.026 and 0.031 respectively).

## Conclusions

Socioeconomic risk factors for persistent inflammation amongst PLWH in SSA differ depending on the type of inflammatory pathway. Understanding these pathways and their socioeconomic drivers will help identify those at risk and target interventions for NCDs. Future studies assessing drivers of inflammation in HIV should include an SES assessment.

## Introduction

Non-communicable diseases (NCDs) are fast becoming the leading cause of mortality in low-income sub-Saharan Africa (SSA), with disability adjusted life years related to NCDs fast approaching those from communicable diseases [1]. Given the increased risk of NCDs amongst people living with HIV (PLWH) in high income countries, there is concern that the ageing HIV population will be particularly at-risk form NCD related morbidity and mortality in SSA, placing further strain on already under-resourced healthcare systems. Persistent inflammation and resultant endothelial damage amongst PLWH are implicated in the patho-genesis of NCDs amongst PLWH across healthcare settings [2–6]. Studies to date identify several drivers of inflammation, including microbial translocation through a compromised gut mucosa, subclinical coinfection, or low-level persistent HIV viraemia [7–9]. Aetiological mechanisms are often complex and overlapping, especially amongst those with a history of advanced immune suppression prior to treatment [10].

We previously demonstrated that heightened inflammation amongst PLWH in Malawi is heterogeneous in nature, with different immunological pathways predicting different trajecto-ries in endothelial damage during the first year of antiretroviral therapy (ART) [5]. Carotid femoral pulse wave velocity (cfPWV) is a gold standard measurement of arterial stiffness and has been validated against clinical outcomes in high-income settings; a cfPWV in the top ver-sus bottom tertile is associated with a >2-fold increased risk of myocardial infarction or stroke [11–13]. The 2012 European Society of Cardiology consensus guidelines propose a 10-m/sec-ond threshold as high risk for CVD events [14].

In this low-income setting, the picture of persistent inflammation amongst PLWH is likely to be further impacted by socio-economic factors such as poor water and housing quality, overcrowded living conditions, low incomes and uncertain food stability leading to increased risk of gastrointestinal infections, malaria, tuberculosis, malnutrition and decreasing ability to present for routine care for acute or chronic illness [15–20].

In order to modify NCD risk factors and reduce clinical burden, a multi-faceted approach involving reduction of traditional cardiovascular risk factors, pragmatic pharmacological strat-egies and public health messages will likely be required. Trials testing pharmacological inter-ventions aimed at reducing inflammation amongst PLWH have so far shown modest effects and it is not clear how current strategies could be translated to low resource settings [21–23]. An understanding of the extent to which socio-economic factors contribute to inflammation

will strengthen the existing model of chronic inflammation in the region and help identify where interventions should be targeted to most effectively reduce non-communicable comorbidity amongst PLWH. Here, we aim to characterise the relationship between socio-economic factors and immune activation in a cohort of PLWH.

## Methods

ART-naïve adults with a new HIV diagnosis and CD4 <100 cells/uL were recruited prospectively from the ART clinic and voluntary HIV testing clinic at Queen Elizabeth Central Hospital, Blantyre, Malawi, (most HIV-positive patients were recruited from the REALITY trial NCT01825031), along with HIV negative adults with no evidence of infection within the previous 3 months. The enrolment visit for HIV-positive participants was 2 weeks after ART initiation to minimise visit burden in this unwell group. The cohort has been previously reported in detail [4]. In brief, participants underwent a detailed clinical assessment and blood draw for markers of immune activation at enrolment and 42 weeks post ART initiation. cfPWV was also carried out at both time points as a measurement of arterial stiffness. Further methods including inter-rater concordance can be found in [4]. Socio-economic evaluation was carried out at the baseline visit using a standardised questionnaire which included domains on housing, assets, household composition, education, employment, finances, transport and health and wellbeing. All participants provided informed written consent and ethical approval was granted by the College of Medicine Research and Ethics Committee (COMREC), University of Malawi (P.09/13/1464) and the University of Liverpool Research and Ethics Committee (UoL000996).

### Characterisation of immune activation

Immune activation was characterised through cell surface immunophenotyping as well as quantification of plasma biomarkers of inflammation, as previously described [4]. Surface immunophenotyping of T-cells was performed on fresh peripheral blood mononuclear cells (PBMCs) using flow cytometry [4]. T-cell activation, exhaustion and senescence was defined as CD38/HLADR, PD1 and CD57 expression, respectively. Monocytes were defined as classical ($CD14^{++}CD16^{-}$), intermediate ($CD14^{++}CD16^{+}$) or nonclassical ($CD14^{+}CD16^{+}$). Stored plasma was tested for 22 cytokines: Proinflammatory Panel-1 (interferon [IFN]-ɣ, interleukin [IL]-1β, IL-2, IL-4, IL-6, IL-8, IL-10, IL12p70, IL-13, tumour necrosis factor [TNF]-α), Vascular Injury Panel-2 (serum amyloid A [SAA], C-reactive protein [CRP], vascular cell adhesion molecule [VCAM]-1, intercellular adhesion molecule [ICAM-1]), Chemokine Panel-1 (macrophage inflammatory protein [MIP]-1β, interferon γ-induced protein [IP]-10, monocyte chemoattractant protein [MCP]-1), Angiogenesis -Panel-1 (vascular endothelial growth factor [VEGF]-A, basic fibroblast growth factor [bFGF]) and single analyte assays for IL-1 receptor antagonist (IL-1Ra) and IL-7., all from Meso Scale Discovery (MSD, Rockland, MD, USA). Assays were performed following the manufacturer's instructions and recommended dilutions for human plasma. Soluble CD163 was measured in plasma diluted at 1:20 using DuoSet antibodies (R&D Systems, Minneapolis, MN, USA) on MSD Multiarray plates. CMV viral loads were quantified by DNA PCR in a subset of participants with available plasma as described previously [24]. Values <100 copies/mL were less than 3 ct values higher than background fluorescence and were therefore considered negative.

### Statistical analysis

All socioeconomic variables were categorical and were compared by HIV status using the chi square tests. Associations were tested for socioeconomic variables and immune activation

markers as well as socioeconomic variables and cfPWV within the HIV cohort. Univariate relationships were tested using Wilcoxon ranksum or Kruskal-Wallis. Socioeconomic variables showing univariate associations with immune activation markers or cfPWV at a p value<0.1 were selected for reporting and entered into multivariate analysis by linear regression. Immune activation markers and cfPWV were log transformed for normality for linear regression analysis and back transformed for presentation of results. Regression models examining associations with socioeconomic variables were adjusted *a priori* for age and sex as confounders for immune activation markers and for age, sex, blood pressure and haemoglobin for cfPWV as previously identified [4]. Adjusted models with p<0.1 were selected for reporting. Analysis was undertaken using Stata 13.1 (StataCorp, College Station, USA).

# Results

## Clinical characteristics

Of 279 PLWH enrolled, the median age was 36 years (IQR 31–43) and 122 (44%) were female. The median (interquartile range (IQR)) nadir CD4 count and HIV viral load were 41 cells/μL (18–62) and 110,000 copies/mL (4000–290,000) respectively. One hundred and ten HIV-negative participants had comparable age (median [IQR] 35 years [31–41]), with 66(60%) women. Forty-five(16%) HIV-positive participants were diagnosed with an acute co-infection at study enrolment and CMV PCR was positive for 61 (32%) of 193 tested HIV-positive participants, with median (IQR) CMV viral load 928 copies/mL (412–3052). The median (IQR) cfPWV for the HIV-positive participants at baseline was 7.3 m/s (6.5–8.2). Fifteen (4%) participants died during the study.

## Socio-economic characteristics

Socio-economic data were available for 267 participants living with HIV and 105 HIV negative participants. Overall, 212 (58%) of participants were renting or living with family and 316 (86%) used a pit latrine toilet. Two hundred and one (55%) had electricity and 95 (26%) had a private water tap at home. Seventy five (21%) were unemployed and of those employed, 108 (29%) earned less than 25 USD per month, although 97 (36%) of participants were not able to provide an estimate of monthly salary. Three hundred and thirty nine (92%) of participants felt that their household did not have enough food.

The main water source for participants differed according to HIV status. Comparing PLWH with participants without HIV, 105 (40%) compared to 18 (17%) used a shared domestic tap and only 59 (22%) compared to 46 (33%) used a water kiosk (p<0.0001, Table 1). There were higher proportions of participants unemployed or working in unskilled labour within the HIV cohort, who were also more likely to work 7 days per week and longer than 8 hours per day, and tended to be on lower salaries (p<0.0001, Table 1). People living with HIV were less likely to grow crops [31 (12%) versus 32 (30%); p<0.0001] and experienced lower rates of food security [13 (5%) vs 16(14%); p<0.0001].

## Socio-economic predictors of cfPWV amongst PLWH

On univariate analysis of the HIV positive cohort, cfPWV was higher for the 16 (8%) participants who travelled to work in a car compared to those who did not [median (IQR) 8m/s (7.3 to 10.2) versus 7.2m/s (6.3 to 8.1); p = 0.008]; for those who owned a television [7.7 (6.7 to 8.5) versus 7.2 (6.2 to 7.8) p = 0.0005]; and for those who had an electricity supply at home [(7.4 (6.5 to 8.3) versus 7.2 (6.4 to 8.1) p = 0.093]. These factors remained associated with cfPWV when adjusted for age, sex, blood pressure and haemoglobin: car ownership [fold change

**Table 1. Socio-economic variables for 367 Malawian adults according to HIV status.**

| Socio-Economic Variables | | | HIV uninfected n = 107* | PLWH n = 266* | p value |
|---|---|---|---|---|---|
| | | | Frequency (%) | Frequency (%) | |
| Housing | Roof | Grass | 7 (7) | 14 (5) | 0.81 |
| | | Corrugated | 96 (91) | 242 (92) | |
| | | Other | 2 (2) | 7 (3) | |
| | Walls | Sundried brick | 48 (46) | 119 (45) | 0.16 |
| | | Burnt brick | 55 (52) | 125 (48) | |
| | | Other | 2 (2) | 18 (7) | |
| | Floor | Earth | 14 (13) | 41 (16) | 0.11 |
| | | Brick | 0 (0) | 8 (3) | |
| | | Cement | 92 (87) | 198 (80) | |
| | Tenure | Bought | 10 (10) | 25 (10) | 0.72 |
| | | Built | 32 (31) | 71 (29) | |
| | | Renting | 52 (51) | 139 (56) | |
| | | Relatives | 8 (8) | 13 (5) | |
| | Toilet | Flush | 17 (16) | 38 (14) | 0.71 |
| | | Latrine | 90 (84) | 226 (86) | |
| | Number of bedrooms | 1 | 12 (11) | 16 (6) | 0.038 |
| | | 2 | 35 (33) | 60 (23) | |
| | | 3 | 29 (27) | 77 (29) | |
| | | 4 | 20 (19) | 65 (24) | |
| | | 5 | 8 (7) | 21 (8) | |
| | | >5 | 3 (3) | 26 (10) | |
| | Kitchen as a separate room | | 64 (62) | 163 (62) | 0.77 |
| Utilities | Cooking | Wood/Charcoal | 94 (90) | 232 (89) | 0.79 |
| | | Electricity/gas | 11 (10) | 31 (12) | |
| | Electricity | | 57 (55) | 144 (55) | 0.91 |
| | Water | Private domestic tap | 31 (30) | 64 (24) | <0.0001 |
| | | Shared domestic tap | 18 (17) | 105 (40) | |
| | | Communal Water kiosk | 46 (44) | 59 (22) | |
| | | Protected well | 8 (8) | 26 (10) | |
| | | Lake/unprotected well | 2 (2) | 10 (4) | |
| Belongings | Fridge | | 41 (38) | 65 (25) | 0.14 |
| | TV | | 40 (37) | 120 (46) | <0.0001 |
| | Video | | 38 (35) | 87 (33) | 0.003 |
| | Radio | | 79 (73) | 163 (62) | 0.26 |
| | Mobile phone | | 87 (81) | 203 (77) | 0.092 |
| | Bicycle | | 19 (18) | 27 (210) | 0.31 |
| | Car | | 5 (5) | 11 (4) | 0.11 |
| Household | Number of adults | 1 | 14 (13) | 43 (16) | 0.87 |
| | | 2 | 56 (52) | 132 (50) | |
| | | 3 | 19 (18) | 50 (19) | |
| | | >3 | 18 (17) | 41 (16) | |
| | Number of children | 0 | 10 (9) | 54 (20) | 0.004 |
| | | 1 | 21 (20) | 44 (17) | |
| | | 2 | 24 (22) | 72 (27) | |
| | | 3 | 21 (20) | 58 (22) | |
| | | >3 | 31 (29) | 38 (14) | |

*(Continued)*

**Table 1.** (Continued)

| Socio-Economic Variables | | | HIV uninfected n = 107* | PLWH n = 266* | p value |
|---|---|---|---|---|---|
| | | | Frequency (%) | Frequency (%) | |
| Education/ employment | Education | None | 5 (6) | 22 (8) | 0.032 |
| | | Primary incomplete | 28 (26) | 88 (34) | |
| | | Primary complete | 10 (9) | 33 (13) | |
| | | Secondary incomplete | 40 (37) | 56 (21) | |
| | | Secondary complete | 16 (15) | 51 (19) | |
| | | Tertiary complete | 9 (8) | 17 (6) | |
| | Occupation | Unemployed | 15 (14) | 60 (23) | <0.0001 |
| | | Student | 24 (22) | 31 (12) | |
| | | Non-skilled labourer | 56 (52) | 141 (53) | |
| | | Skilled labourer | 5 (5) | 27 (10) | |
| | | Professional worker | 8 (7) | 7 (3) | |
| | Employment status | Full time | 68 (79) | 123 (66) | 0.12 |
| | | Part time | 4 (5) | 21 (11) | |
| | | Not working—ill health | 2 (2) | 10 (5) | |
| | | Not working—lack of employment | 12 (14) | 32 (17) | |
| | Working days per week | 4 or less | 3 (5) | 10 (10) | <0.0001 |
| | | 5 | 31 (48) | 18 (17) | |
| | | 6 | 26 (240) | 45 (43) | |
| | | 7 | 5 (8) | 32 (30) | |
| | Working hours per day | Less than 8 | 13 (18) | 19 (16) | <0.0001 |
| | | 8 | 48 (66) | 33 (27) | |
| | | 9 to 12 | 12 (16) | 58 (48) | |
| | | More than 12 | 0 (0) | 12 (10) | |
| Finances | Paid by Salary | | 55 (75) | 78 (57) | 0.007 |
| | Participant income (USD/month) | <10 | 1 (1) | 63 (37) | <0.0001 |
| | | 10–25 | 15 (18) | 29 (17) | |
| | | 26–50 | 34 (40) | 31 (18) | |
| | | 51–100 | 21 (25) | 25 (15) | |
| | | >100 | 13 (15) | 23 (13) | |
| | Household income (USD / month) | <10 | 1 (1) | 61 (36) | <0.0001 |
| | | 10–25 | 6 (8) | 23 (13) | |
| | | 26–50 | 31 (39) | 26 (15) | |
| | | 51–100 | 18 (23) | 31 (18) | |
| | | >100 | 24 (30) | 30 (18) | |
| | In receipt of benefits | | 0 (0) | 27 (15) | <0.0001 |
| | In receipt of household credit | | 76 (70) | 93 (37) | <0.0001 |
| | Source of credit | Family | 1 (1) | 19 (22) | <0.0001 |
| | | Friends | 40 (52) | 33 (38) | |
| | | Bank | 8 (10) | 16 (18) | |
| | | Co-operative | 28 (36) | 20 (23) | |
| | Household has enough food | | 16 (14) | 13 (5) | <0.0001 |
| | Household grows crops | | 32 (30) | 31 (12) | <0.0001 |
| | Households with a child that has left school due to lack of food | | 1 (1) | 8 (3) | 0.22 |

(*Continued*)

**Table 1.** (Continued)

| Socio-Economic Variables | | | HIV uninfected n = 107[*] | PLWH n = 266[*] | p value |
|---|---|---|---|---|---|
| | | | Frequency (%) | Frequency (%) | |
| Transport to work | Mode of transport | Walking | 16 (16) | 32 (12) | 0.66 |
| | | Bus | 85 (83) | 217 (84) | |
| | | Car | 1 (1) | 7 (3) | |
| | | Other | 1 (1) | 2 (1) | |
| | Transport costs (Malawian Kwacha) | <200 | 2 (2) | 2 (1) | 0.047 |
| | | 200–350 | 33 (37) | 101 (46) | |
| | | 350–600 | 27 (30) | 37 (17) | |
| | | 600–1000 | 15 (17) | 30 (14) | |
| | | >1000 | 13 (14) | 48 (22) | |
| | Travel time | <10 | 2 (2) | 6 (2) | <0.0001 |
| | | 10–30 | 23 (22) | 15 (6) | |
| | | 31–60 | 45 (43) | 91 (38) | |
| | | 61–120 | 30 (29) | 110 (46) | |
| | | >120 | 4 (4) | 19 (8) | |
| Health and Wellbeing | Health visual analogue score | 0 | 31 (29) | 77 (30) | 0.094 |
| | | 1 | 46 (43) | 82 (32) | |
| | | 2 | 25 (23) | 60 (23) | |
| | | 3 | 5 (5) | 23 (9) | |
| | | 4 | 0 (0) | 12 (5) | |
| | | 5 | 1 (1) | 2 (1) | |
| | Wellbeing visual analogue score | 0 | 27 (25) | 80 (31) | 0.042 |
| | | 1 | 52 (48) | 88 (35) | |
| | | 2 | 25 (23) | 57 (22) | |
| | | 3 | 2 (2) | 21 (8) | |
| | | 4 | 1 (1) | 8 (3) | |
| | | 5 | 1 (1) | 1 (0.4) | |

[*]Maximum number of SES questionnaires completed.

Some categories have missing values and total number of available answers for each category is inferred from percentage denominator. Denominator for employment indices were those eligible for employment and did not include students. PLWH = People living with HIV

1.3m/s (95% CI 1.10 to 1.56); p = 0.003]; television ownership [1.12m/s (1.03 to 1.23); p = 0.012]; and electricity access [1.09m/s (1.01 to 1.17); p = 0.029]. Median values for plasma biomarkers are shown for each socio-economic variables and further detail on adjusted models are reported in S1 File.

## Socio-economic predictors of immune activation amongst PLWH

**Education level and income.** Amongst the HIV positive cohort, increasing education level and income were associated with increasing proportion of activated CD8 T cells (Fig 1). Median (IQR) proportion of activated CD8 T cells ranged from 70% (63–78) amongst those with no education to 88% (74–94) amongst those with a tertiary education (p = 0.002); and 71% (55–81) amongst those earning less than 10 USD/month to 87% (86–93) amongst those earning between 100 to 150 USD/month(p = 0.0001). Interestingly, an inverse association was observed with intermediate monocytes, with higher median (IQR) amongst those with no

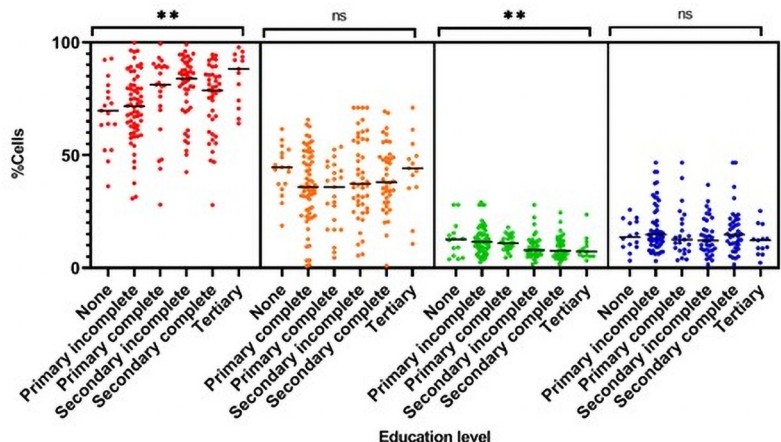

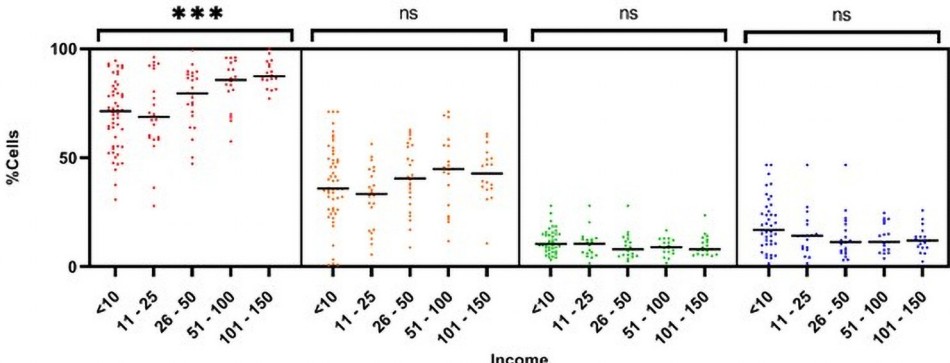

**Fig 1. Cell surface immune activation markers according to education and income category.** *p<0.01 **p>0.001 ***p<0.0001 ns p>0.01.

education compared to a tertiary education [12.6% (5.4–15.5) versus 7.3% (5.3–9.8); p = 0.01] and amongst those earning less than 10 USD/month compared to 100–150 USD/month [10.5% (7.8–15.0) versus 8.0% (5.0–12.9); p = 0.08].

Participants with a tertiary education tended to experience greater improvements in CD8 activation after 42 weeks of ART compared to those without any education [%median (IQR) change from baseline to 42 weeks -5.8 (-16.7 to 2.2) versus 2.4 (-7.9 to 12); p = 0.07], as did those with higher incomes [% change 6.0 (-8.2 to 20.1) versus -18.2 (-21.1 to -4.8) comparing lowest and highest income categories; p value = 0.0001]. However, neither educational level nor patient income predicted change in the proportion of intermediate monocytes on ART (p = 0.69 and 0.41 respectively).

Adjusting for age and sex, we first compared those with some education to those without any: IL7 was lower for those with some education [fold change 0.20µg/mL (0.07 to 0.59); p = 0.002] who also trended towards lower bFGF [fold change 0.22µg/mL (0.048 to 1.09);

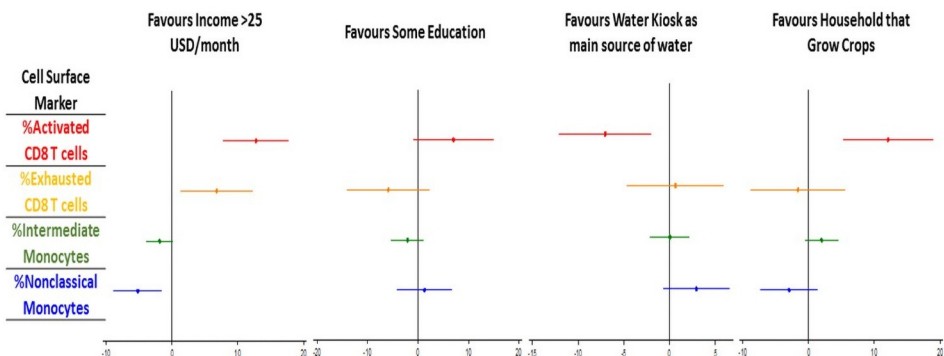

**Fig 2. Adjusted fold change in cell surface immune activation markers according to socio-economic risk factors.**
Models for the effect of socio-economic variables on CD8 T cell and monocyte phenotypes, adjusted for age and sex, are shown. The x axis shows fold change with 95% confidence intervals for the following socio-economic comparisons: i) Income >25 USD per month compared to income</ = 25USD/month ii) some education compared to no education iii) water kiosk as source of water compared to all other water sources iv) grows household crops compared to doesn't grow household crops.

p = 0.056] and higher proportion of activated CD8 T cells [fold change 7.02% (-0.97 to 15.0); p = 0.085] (Figs 2 and 3). Amongst those earning more than 25 USD/month compared to those earning 25 USD/month or less, adjusted CD8 activation and exhaustion were both higher [activation fold change 12.7% (95% CI 7.75 to 17.78), p<0.0001; exhaustion 6.77% (1.30 to 12.24), p = 0.016]. Levels of IL6 and IL13 were also higher amongst people with higher incomes [fold change (95% CI) 2.9 µg/mL (1.13 to 7.6), p = 0.028; and 3.1 µg/mL (1.2 to 8.3), p = 0.025 respectively]. Proportions of nonclassical monocytes were lower amongst this higher income bracket with lower intermediate monocytes trending towards significance [nonclassical fold change -5.23 (95%CI -8.94 to -1.53), p = 0.006; intermediate -1.91 (-3.91 to 0.10), p = 0.063].

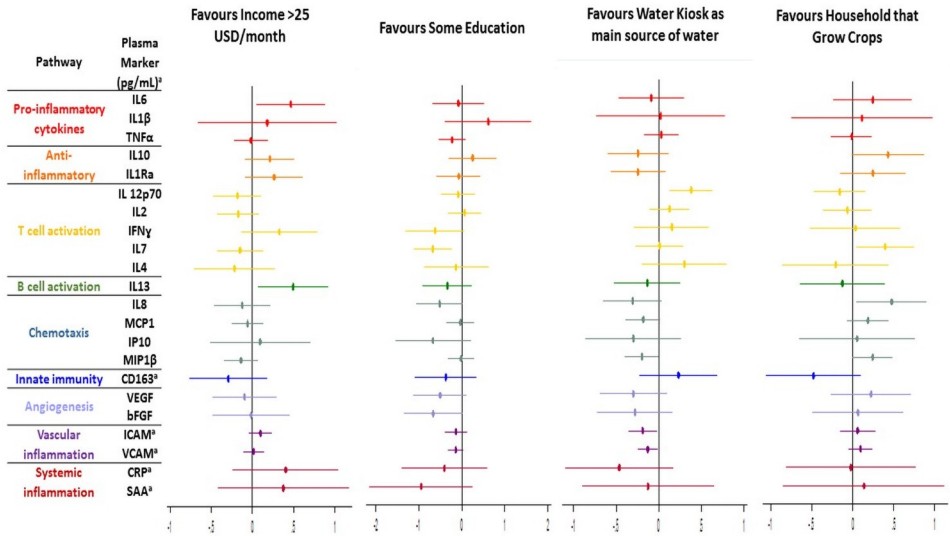

**Fig 3. Adjusted fold change in plasma inflammatory markers according to socio-economic risk factors.** All inflammatory biomarkers measured in µg/mL apart from those marked * which are in pg/mL.

**Household factors.** Participants with HIV who grew crops at home had higher CD8 activation [median (IQR) 88% (80–91) versus 76% (64–86); p = 0.0004], and experienced greater improvements in CD8 activation after 42 weeks of ART [median (IQR) -15.8% (-19.9 to -1.9) versus -0.18 (-13.1 to 12.1); p = 0.01] over 42 weeks of ART. Baseline CD8 activation remained significantly higher amongst those who grew crops in adjusted analysis [fold change 12.1% (5.2 to 19.0);p = 0.001], with significantly higher adjusted levels of IL7 and IL8 [IL7 fold change 2.47pg/mL (95%CI 1.10 to 5.52),p = 0.028; IL8 2.97pg/mL (1.10 to 7.97),p = 0.031].

Compared to those with a brick floor, participants with an earth floor had higher proportions of nonclassical monocytes at presentation [mean (IQR) 16.9% (9.9 to 25.80) versus 10.3% (8.2–16.7); p = 0.04] and were also more likely to experience improvements after 42 weeks of ART [nonclassical monocyte mean (IQR) change -2.2% (-12.0 to 2.9) versus 5.8% (3.9 to 7.2); p = 0.026].

**Water source.** Amongst PLWH using a water kiosk compared to other water sources, the median (IQR) proportion of activated CD8 T cells at baseline was lower [mean (IQR) 70% (59 to 81) versus 81% (68 to 89), p = 0.002], and decreases in CD8 T cell activation after 42 weeks of ART were less marked [-3.78 (-15.79 to 7.18) versus -9.85 (-9.80 to 20.7); p = 0.039]. Water kiosk use was also associated with lower rates of CMV PCR positivity [5(7%) versus 57(31%); p<0.0001]. Use of a shared domestic tap was overrepresented as the main water source amongst those with CMV (shared domestic tap users compared to water kiosk; 32 (86%) versus 5 (14%); p<0.0001). For the four PLWH who used an unprotected well as their main water source, the proportion of intermediate monocytes was higher than those who did not [median % 14.2 (95%CI 11.5–17.6) versus 9.3 (5.7–13.0), p = 0.068].

After adjustment for confounders, CD8 activation remained lower amongst water kiosk users [fold change -7.05% (95%CI -12.1 to -1.97); p = 0.007]. MIP1β, sICAM1 and sVCAM1 were also all significantly lower amongst kiosk users [fold change 0.63pg/mL (95%CI 0.40 to 0.99), p = 0.047; 0.65μg/mL (0.44 to 0.59), p = 0.026; and 0.74 μg/mL (0.57 to 0.97), p = 0.031 respectively] but IL12p70 was higher [2.39 (1.34 to 4.28); p = 0.003].

## Discussion

We show that, amongst PLWH in a low income SSA setting, socioeconomic variables associate with chronic immune activation along two different inflammatory pathways. Activated CD8 T cells were expanded amongst those with variables consistent with a higher socioeconomic status including higher incomes, higher education and having home grown crops. This population was also more likely to show improvements in CD8 T cell activation on ART. Conversely CD16 positive monocytes, both non-classical and intermediate, were expanded amongst those with variables consistent with a lower socio-economic status including lower income, less education and earth flooring. This expansion of CD16 positive monocytes did not resolve on ART.

The observation that participants in higher socioeconomic groups were more likely to improve on ART has several possible explanations. Firstly, inflammation in this group may be related to HIV itself, or viral co-infection, resulting in T cell activation. Secondly, there may be an element of the ART care effect, whereby other traditional risk factors are managed once engaged in care. Lastly, although the socioeconomic variables associated with arterial stiffness (car and TV ownership, electricity connection) were not the same as those associated with T cell activation, they were consistent with a higher socio-economic status and perhaps a more sedentary lifestyle. Taken together with previous data showing that T cell exhaustion is linked with arterial stiffness during the first 3 months of anti-retroviral therapy [4], there is a suggestion that there may be a relationship between T cell activation, sedentary lifestyles and

increased cardiovascular risk amongst PLWH in low income SSA setting. The observation that T cell activation may be increased amongst PLWH of higher socioeconomic status has not been previously described. However, T cell activation, exhaustion and senescence have been previously linked to premature ageing and sedentary lifestyles [25, 26]. Studies for older populations show that lower levels of physical activity predispose to a shift towards Th2 responses and T cell senescence [26]. Non-communicable disease risk factors have previously been shown to vary across socio-economic backgrounds in sub-Saharan Africa, but understanding of this relationship is limited by heterogeneity in measurement and reporting of socio-economic status [27]. Urbanisation and epidemiological transition are likely to favour different non-communicable disease risks depending on socio-economic background; for example, sedentary lifestyle being more prominent in high income groups and poor diet in low-income groups [27]. Detailed characterisation of this relationship will be needed in future studies to provide context for the role of inflammation within these models.

Participants in lower socio-economic groups experience expanded CD16+ monocytes, which did not improve on ART. This suggests that this aspect of innate immune activation is not linked to HIV itself or co-infections that would normally improve on ART. Subclinical TB, untreated or recurrent acute bacterial and malarial infections, or malnutrition with persistent microbial translocation could play a role. Although this association has not been previously reported amongst PLWH, there is evidence to suggest that factors associated with lower socio-economic status are related to immune activation in other fields. In paediatrics, lower parental socioeconomic status is associated with higher levels of CRP and enhancing nutrition can improve the effects of inflammation [28, 29]. A socioeconomic assessment of the Framingham Offspring Study showed that both higher lifetime socioeconomic status and education level predicted lower levels of inflammatory cytokines [30, 31]. Chronic life stressors might also contribute to higher inflammation levels amongst this more economically vulnerable group [32].

Water kiosk use was independently associated with lower levels of CD8 activation, inflammatory and endothelial cytokines and CMV viraemia. Water kiosks sell water in containers from a centralised source. It is possible that use may therefore reflect a lower infection-driven inflammatory risk profile due to improved water quality or better sanitation environments. Sima and colleagues in Jakatar showed that water kiosks produced good quality water and reduced diarrhoeal illness to a similar extent to bottled water when compared with using wells as a water source [33]. Shared domestic pipe use was more common among those who tested CMV PCR positive, perhaps reflecting overcrowding in urban settings leading to increased risk of close contact transmission. Intermediate monocytes were expanded amongst four PLWH who used an unprotected well as their main water source. Although this group is very small, this finding is in keeping with low socio-economic status and unsafe water sources being associated with inflammatory monocyte phenotypes and innate inflammatory pathways [34].

Study strengths include careful characterisation of socio-economic status in a longitudinal cohort study of clinically evaluated participants with and without HIV. We compare socioeconomic data with cell surface immunophenotyping and a comprehensive panel of inflammatory biomarkers allowing a biosocial assessment of chronic inflammation among PLWH in low income SSA.

Because our cohort of PLWH all had advanced immune suppression, these findings may not be generalisable to HIV populations as a whole. Also, there may be differences in reasons for late presentation to care amongst those from higher and lower socio-economic status backgrounds such as cultural pressures or ability to take time off work. The sample size for this study may not have allowed us to identify smaller but important effects of socioeconomic

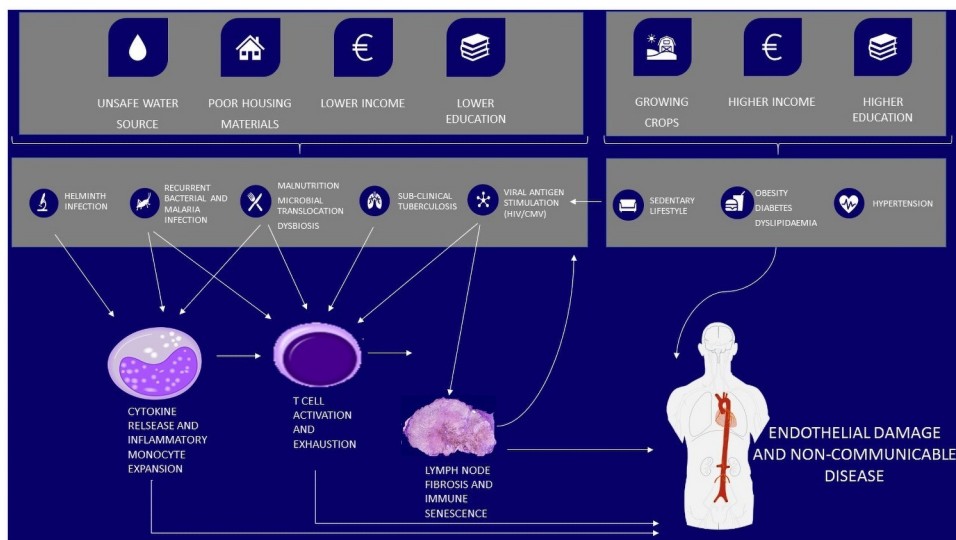

**Fig 4. Hypothesis for the role of socio-economic determinants in chronic inflammation and endothelial damage amongst PLWH in low income SSA.** Hypothesis for the impact of socio-economic factors on inflammation mediated non-communicable diseases in people living with HIV in low-income settings. This builds on previously documented relationships between drivers of inflammation, and its effect on endothelial damage in this setting [4, 5]. Further research will be required to evaluate and improve our understanding of the factors driving immune activation and non-communicable disease in low-income settings.

variables on inflammation and arterial stiffness. Analysis of a large number of inflammatory and socio-economic markers may have led to an increase in type 1 error, and although we have presented results in keeping with an *a priori* identified biologically plausible hypothesis, it is important to note that individual mechanisms will require further characterisation in larger clinical studies.

Researchers are increasingly recognising that immune activation in PLWH is a heterogenous process and interest is emerging around discrete clinical inflammatory phenotypes [5]. Here we show that socioeconomic risk factors for inflammation amongst PLWH have varied effects (summarised in Fig 4). Understanding the extent to which socioeconomic factors contribute to such phenotypes, especially in low income SSA, will help identify those most at risk of developing inflammation driven non-communicable disease, support policy to reduce non-communicable disease risk factors and aid efficient allocation of limited health care resources. Future studies assessing drivers of inflammation amongst PLWH should include an assessment of socioeconomic status.

## Supporting information

**S1 File.**
(DOCX)

## Acknowledgments

The authors would like to acknowledge the patients and their families as well as the staff in the ART clinic and Department of Medicine at Queen Elizabeth Central Hospital, Blantyre, Malawi. We would also like to acknowledge Dr Elizabeth Tilley of the Malawi Polytechnic, for support with interpreting water supply data.

We would like to thank the REALITY trial group for their support with study design and implementation.

The REALITY trial group consists of:

Participating Centres: Joint Clinical Research Centre (JCRC), Kampala, Uganda (coordinating centre for Uganda): P Mugyenyi, C Kityo, V Musiime, P Wavamunno, E Nambi, P Ocitti, M Ndigendawani. JCRC, Fort Portal, Uganda: S Kabahenda, M Kemigisa, J Acen, D Olebo, G Mpamize, A Amone, D Okweny, A Mbonye, F Nambaziira, A Rweyora, M Kangah and V Kabaswahili. JCRC, Gulu, Uganda: J Abach, G Abongomera, J Omongin, I Aciro, A Philliam, B Arach, E Ocung, G Amone, P Miles, C Adong, C Tumsuiime, P Kidega, B Otto, F Apio. JCRC, Mbale, Uganda: K Baleeta, A Mukuye, M Abwola, F Ssennono, D Baliruno, S Tuhirwe, R Namisi, F Kigongo, D Kikyonkyo, F Mushahara, D Okweny, J Tusiime, A Musiime, A Nankya, D Atwongyeire, S Sirikye, S Mula, N Noowe. JCRC, Mbarara, Uganda: A Lugemwa, M Kasozi, S Mwebe, L Atwine, T Senkindu, T Natuhurira, C Katemba, E Ninsiima, M Acaku J Kyomuhangi, R Ankunda, D Tukwasibwe, L Ayesiga. University of Zimbabwe Clinical Research Centre, Harare, Zimbabwe: J Hakim, K Nathoo, M Bwakura-Dangarembizi, A Reid, E Chidziva, T Mhute, GC Tinago, J Bhiri, S Mudzingwa, M Phiri, J Steamer, R Nhema, C Warambwa, G Musoro, S Mutsai, B Nemasango, C Moyo, S Chitongo, K Rashirai, S Vhembo, B Mlambo, S Nkomani, B Ndemera, M Willard, C Berejena, Y Musodza, P Matiza, B Mudenge, V Guti. KEMRI Wellcome Trust Research Programme, Kilifi, Kenya: A Etyang, C Agutu, J Berkley, K Maitland, P Njuguna, S Mwaringa, T Etyang, K Awuondo, S Wale, J Shangala, J Kithunga, S Mwarumba, S Said Maitha, R Mutai, M Lozi Lewa, G Mwambingu, A Mwanzu, C Kalama, H Latham, J Shikuku, A Fondo, A Njogu, C Khadenge, B Mwakisha. Moi University Clinical Research Centre, Eldoret, Kenya: A Siika, K Wools-Kaloustian, W Nyandiko, P Cheruiyot, A Sudoi, S Wachira, B Meli, M Karoney, A Nzioka, M Tanui, M Mokaya, W Ekiru, C Mboya, D Mwimali, C Mengich, J Choge, W Injera, K Njenga, S Cherutich, M Anyango Orido, G Omondi Lwande, P Rutto, A Mudogo, I Kutto, A Shali, L Jaika, H Jerotich, M Pierre. Department of Medicine and Malawi-Liverpool Wellcome Trust Clinical Research Programme, College of Medicine, Blantyre, Malawi: J Mallewa, S Kaunda, J Van Oosterhout, B O'Hare, R Heyderman, C Gonzalez, N Dzabala, C Kelly, B Denis, G Selemani, L Nyondo Mipando, E Chirwa, P Banda, L Mvula, H Msuku, M Ziwoya, Y Manda, S Nicholas, C Masesa, T Mwalukomo, L Makhaza, I Sheha, J Bwanali, M Limbuni. Trial Coordination and Oversight: MRC Clinical Trials Unit at UCL, London, UK: D Gibb, M Thomason, AS Walker, S Pett, A Szubert, A Griffiths, H Wilkes, C Rajapakse, M Spyer, A Prendergast, N Klein. Data Management Systems: M Rauchenberger, N Van Looy, E Little, K Fairbrother. Social Science Group: F Cowan, J Seeley, S Bernays, R Kawuma, Z Mupambireyi.

Independent REALITY Trial Monitors: F Kyomuhendo, S Nakalanzi, J Peshu, S Ndaa, J Chabuka, N Mkandawire, L Matandika, C Kapuya. Trial Steering Committee: I Weller (Chair), E Malianga, C Mwansambo, F Miiro, P Elyanu, E Bukusi, E Katabira, O Mugurungi, D Gibb, J Hakim, A Etyang, P Mugyenyi, J Mallewa. Data Monitoring Committee: T Peto (Chair), P Musoke, J Matenga, S Phiri.

Endpoint Review Committee (independent members): H Lyall (Co-Chair), V Johnston (Co-Chair), F Fitzgerald, F Post, F Ssali, A Prendergast, A Arenas-Pinto, A Turkova, A Bamford.

## Author Contributions

**Conceptualization:** Christine Kelly, Irene Sheha, Mishek Chammudzi, Kondwani Jambo, Jane Mallewa, Alicja Rapala, Patrick W. G. Mallon, Henry Mwandumba, Nigel Klein, Saye Khoo.

**Data curation:** Christine Kelly, Patricia Hunter, Alejandro Garcia Abner, Raphael Kamn'gona, Irene Sheha, Mishek Chammudzi, Kondwani Jambo, Alicja Rapala, Henry Mwandumba, Saye Khoo.

**Formal analysis:** Christine Kelly, Willard Tinago, Kondwani Jambo, Alicja Rapala, Patrick W. G. Mallon, Saye Khoo.

**Funding acquisition:** Christine Kelly, Jane Mallewa, Henry Mwandumba, Nigel Klein, Saye Khoo.

**Investigation:** Christine Kelly, Dagmar Alber, Patricia Hunter, Natasha Luckhurst, Jake Connolly, Francesca Arrigoni, Raphael Kamn'gona, Irene Sheha, Mishek Chammudzi, Kondwani Jambo, Alicja Rapala, Henry Mwandumba, Nigel Klein, Saye Khoo.

**Methodology:** Christine Kelly, Willard Tinago, Patricia Hunter, Alejandro Garcia Abner, Raphael Kamn'gona, Irene Sheha, Mishek Chammudzi, Kondwani Jambo, Jane Mallewa, Alicja Rapala, Patrick W. G. Mallon, Henry Mwandumba, Nigel Klein, Saye Khoo.

**Project administration:** Christine Kelly, Raphael Kamn'gona, Irene Sheha, Mishek Chammudzi, Jane Mallewa, Saye Khoo.

**Resources:** Christine Kelly, Patrick W. G. Mallon, Henry Mwandumba, Nigel Klein, Saye Khoo.

**Supervision:** Christine Kelly, Kondwani Jambo, Saye Khoo.

**Validation:** Kondwani Jambo.

**Writing – original draft:** Christine Kelly, Willard Tinago.

**Writing – review & editing:** Christine Kelly, Willard Tinago, Dagmar Alber, Patricia Hunter, Natasha Luckhurst, Jake Connolly, Francesca Arrigoni, Alejandro Garcia Abner, Raphael Kamn'gona, Irene Sheha, Mishek Chammudzi, Kondwani Jambo, Jane Mallewa, Alicja Rapala, Patrick W. G. Mallon, Henry Mwandumba, Nigel Klein, Saye Khoo.

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
