## [Decision Letter · Decision Letter 0]

19 Mar 2021

PONE-D-20-21907

Inflammatory pathways amongst people living with HIV in Malawi differ according to socioeconomic status

PLOS ONE

Dear Dr. Kelly,

Thank you for submitting your manuscript to PLOS ONE. After careful consideration, we feel that it has merit but does not fully meet PLOS ONE’s publication criteria as it currently stands. Therefore, we invite you to submit a revised version of the manuscript that addresses the points raised during the review process.

We look forward to receiving your revised manuscript.

Kind regards,

Olalekan Uthman, MD, MPH, PhD, FRSPH, FHEA

Academic Editor

PLOS ONE

Journal Requirements

The funders had no role in study design, data collection and analysis, decision to publish, or preparation of the manuscript

Reviewers' comments:

Reviewer's Responses to Questions

**Comments to the Author**

1. Is the manuscript technically sound, and do the data support the conclusions?

Reviewer #1: Partly

Reviewer #2: Yes

2. Has the statistical analysis been performed appropriately and rigorously? 

Reviewer #1: Yes

Reviewer #2: Yes

3. Have the authors made all data underlying the findings in their manuscript fully available?

Reviewer #1: Yes

Reviewer #2: Yes

4. Is the manuscript presented in an intelligible fashion and written in standard English?

Reviewer #1: Yes

Reviewer #2: Yes

5. Review Comments to the Author

Reviewer #1: This is an interesting study which is one of the earliest to shed some light on socio-economic factors which contribute to inflammation amongst PLWH with the view to inform the management of non-communicable diseases. While the results are of importance to the field, there are many aspects that have to be addressed.

Abstract:

“Results: Of 279 PLWH, the median (IQR) age was 36 (31-43) years and 122 (44%) female” – should this be were female?

“and amongst those earning less than 10 USD/month compared to 100-150 USD/month (10.5% versus 8.0%; p=0.08)” I am not sure why this finding is specifically mentioned in the abstract since it was not statistically significant.

Line 45-7: “Non-communicable diseases (NCDs) are fast becoming the leading cause of mortality in low income sub-Saharan Africa (SSA), with DALYs related to NCDs fast approaching those from including communicable diseases(1).” – Please rephrase for clarity: what does from including communicable diseases mean?

Line 50-1: “Persistent inflammation and resultant endothelial damage amongst PLWH is implicated…” – This should be plural.

Some abbreviations are not explained e.g. DALYs, TB, PBMCs, IQR.

Line 89: Was cfPWV performed by a trained person? Was it performed by a single person? If not, was inter-rater concordance assessed?

Line 90: When was the socio-economic evaluation done? If done at both time points, which values were used?

Line 91: Has the standardised questionnaire for assessment of socio-economic status been validated for the sub-Saharan African setting?

Line 97-103: No information is provided about the instrument used for flow cytometry, the antibody manufacturer, the T-cell subsets examined, or the gating strategy used.

Line 104: “Proinflammatory Panel-1 (IFN-Ɣ, IL-1β, IL-2, IL-4, IL-6, IL-8, IL-10, IL12p70, IL-13, TNF-α)” – IL1Ra, IL-10 and IL-13 are anti-inflammatory markers.

Line 101-6: It would be better to write all the biomarker names out in full the first time they are used, since the abbreviations may not be familiar to all the readership of this journal.

Line 110-1: “CMV viral loads were quantified by DNA PCR in a subset of participants with available plasma as described previously(20).” Please explain who this subset was or what a decision to test for CMV was based on.

Write numbers in the beginning of a sentence out in full.

Line 137: Were HIV-uninfected participants also screened for CMV infection?

Line 138: Was cfPWV only performed for the HIV-positive participants?

Line 151: “HIV cohort, who were also more likely to work 7 hours per week…” – should this be 7 days per week?

Table 1: Please check all numbers in Table 1. For instance, the categories for the floor variable do not add up to 100% for the HIV-infected participants. In addition, 198/262 does not equal 80%; 71/262 does not equal 29%; 139/262 does not equal 56%; 90/105 does not equal 84%; 52/105 does not equal 51%; 64/105 does not equal 62%; etc, etc. Note that dark lines that divides the “mode of transport” category.

Line 158-9: “cfPWV was higher for the 16 (8%) participants who travelled to work in a car compared to those who did not [8m/s (7.3 to 10.2)”. It is not clear what is being presented: median and interquartile range? This is also true later in the manuscript e.g. when monocyte data are presented.

Line 161: “p0.0005” should be p=0.0005.

Line 162: “These factors remained predictive of cfPWV”. It seems that associated is the more appropriate word here since it does not appear as if predictive models had been constructed.

If “Carotid femoral pulse wave velocity (cfPWV) was also carried out at both time points as a

measurement of arterial stiffness”, which values are presented in lines 137 and 138, and which were used in the section “Socio-economic predictors of cfPWV amongst PLWH”? Had there been any change in cfPWV over time?

Since many readers will not be familiar with cfPWV, there should be some explanation about what higher values mean in terms of arterial stiffness.

Line 174: What is meant by “an indirect correlation”? According to the statistics, no correlations had been performed.

Line 181: “as did those on higher incomes” – should this be with higher incomes?

Lines 185-195: What is meant by the fold change values? Is this the change over time? Why was this selected rather than the median values for each time point? This is not explained in the statistical analysis section.

Line 191: Since IL-6 has been reported to be affected by obesity, was the association with higher income assessed for confounding with body mass index?

Line 191-3: “Levels of IL6 and IL13 were also higher amongst people with higher incomes [2.9 μg/mL (1.13 to 7.6), p=0.028; and 3.1 μg/mL (1.2 to 8.3), p=0.025 respectively].” Why are the median values (I presume these are medians and IQRs?) shown here and not the fold change? Which time point is referred to? Why are the comparator values not shown for participants with lower incomes? Which groups were combined to create the group “higher incomes”?

Line 193: What is meant by “Adjusted nonclassical and intermediate monocytes”?

Line 211-4: “Baseline CD8 activation remained significantly higher amongst those who grew crops in adjusted analysis [fold change 12.1% (5.2 to 213 19.0);p=0.001], with significantly higher adjusted levels of IL7 and IL8 [IL7 fold change 214 2.47pg/mL (95%CI 1.10 to 5.52),p=0.028; IL8 2.97pg/mL (1.10 to 7.97),p=0.031].” Where are comparator values for participants who did not grow crops? The same is true for lines 231-5.

A table with the descriptive results, as well as the unadjusted and adjusted results of the univariate and multivariable analysis of all the biomarkers tested would be very helpful and greatly improve understanding of the results. One would like to see the results of all the markers that had been tested. In addition, results should be displayed in a systematic fashion since now only results that the authors seemingly found interesting are displayed, even in the supplementary tables. This makes a full assessment of the results impossible.

Line 231: “After adjustment for confounders…”. It is not clear what confounding had been adjusted for with regards to CD8 activation.

Had any adjustment been made for CMV co-infection?

Monocytes were only defined as classical (CD14++CD16-), intermediate (CD14++CD16+) or nonclassical (CD14+CD16+) and no markers of activation had been included in the flow cytometric analysis. For this reason, any statement about monocyte activation is incorrect and should be rephrased and re-interpreted according to the phenotype observed. Just two examples where this is problematic are the following: “population did not show improvements in monocyte activation on ART.”; “Intermediate monocytes were expanded amongst four PLWH who used an unprotected well as their main water source. Although this group is very small, this finding is in keeping with low socio-economic status and unsafe water sources being associated with monocyte activation and innate inflammatory pathways(29).” As it stands, it is completely unclear what it means to have a higher proportion of intermediate monocytes, for instance.

It is not clear what the purpose of the control group is, since only socio-economic variables were compared between them and the HIV-infected participants.

Figure 1: there is no indication of what the asterisks mean.

Figure 2 is very difficult to understand and should be augmented with a footnote that describes what is depicted.

Figure 4 would be greatly strengthened by the inclusion of supporting references. It is not clear how viral antigen stimulation leads to lymph node fibrosis.

Why were CD4 cell markers not also assessed? Why are the data on T cell exhaustion and senescence not also shown?

Reviewer #2: In this paper, Kelly et al build upon previous published work describing distinct immunophenotypes in PLWHIV with low CD4 counts by examining the role of socio-economic status. This is a very important and frequently neglected question. It is particularly poorly studied in the setting of low income countries and I commend the authors on examining this question. Overall, the manuscript well-written and clear, with use of appropriate methodology. The authprs characterise SES and immune activation using multiple measures and report that PLWHIV of higher SES have higher proportions of activated CD8+ T-cells and higher cfPWV. These changes normalise with ART. In contrast, PLWHIV of lower SES have higher proprtions of intermediate monocytes, and do not experience a normalisation of this or a change in T-cell activation with ART. The work contributes to what is known on this topic, and the discussion sites the work in existing literature and appropriately presents limitations of the study. I have a few very minor suggestions but would recommend that the paper be accepted.

• What is a water kiosk? The authors report that it provides clean water but I would like a little more information on what it is.

• In line 151, should 7 hours be 7 days per week

• The authors could include a brief comment on what is known on the effect of repeated social stress on myelopoiesis (e.g. www.pnas.org/content/110/41/16574)

• I think a paragraph on what is known on SES and NCDs in low-income countries would also benefit the paper – e.g. referencing this article amongst others https://www.thelancet.com/journals/langlo/article/PIIS2214-109X(17)30054-2/fulltext

6. PLOS authors have the option to publish the peer review history of their article (what does this mean?). If published, this will include your full peer review and any attached files.

Reviewer #1: No

Reviewer #2: No

---

## [Author Response · Author response to Decision Letter 0]

13 May 2021

Reviewer #1 

Abstract:

“Results: Of 279 PLWH, the median (IQR) age was 36 (31-43) years and 122 (44%) female” – should this be were female?

Correction added to manuscript

“and amongst those earning less than 10 USD/month compared to 100-150 USD/month (10.5% versus 8.0%; p=0.08)” I am not sure why this finding is specifically mentioned in the abstract since it was not statistically significant.

Thank you for highlighting this inaccuracy. Correction added to manuscript. 

Line 45-7: “Non-communicable diseases (NCDs) are fast becoming the leading cause of mortality in low income sub-Saharan Africa (SSA), with DALYs related to NCDs fast approaching those from including communicable diseases(1).” – Please rephrase for clarity: what does from including communicable diseases mean?

Thank you for highlighting this typographical error. Corrected in manuscript.

Line 50-1: “Persistent inflammation and resultant endothelial damage amongst PLWH is implicated…” – This should be plural.

Corrected in manuscript.

Some abbreviations are not explained e.g. DALYs, TB, PBMCs, IQR.

Abbreviations now explained in the manuscript.

Line 89: Was cfPWV performed by a trained person? Was it performed by a single person? If not, was inter-rater concordance assessed?

Full training for cfPWV was provided by colleagues at the Institute of Cardiovascular Sciences, UCL (included on authorship). A sentence has been added to the manuscript to direct readers to a reference with a more detailed outline of how cfPWV was conducted for this cohort. 

Line 90: When was the socio-economic evaluation done? If done at both time points, which values were used?

The assessment was carried out at baseline visit – this has been clarified in the manuscript. 

Line 91: Has the standardised questionnaire for assessment of socio-economic status been validated for the sub-Saharan African setting?

The questionnaire has been used in HIV clinical trials across low-income sub-Saharan Africa, and was initially designed for use in the DART trial (Routine versus clinically driven laboratory monitoring of HIV antiretroviral therapy in Africa (DART): a randomised non-inferiority trial. January 2010. DOI: 10.1016/S0140-6736(09)62067-5). 

Line 97-103: No information is provided about the instrument used for flow cytometry, the antibody manufacturer, the T-cell subsets examined, or the gating strategy used.

A reference has been inserted to refer the reader to the immunophenotyping protocol previously published. 

Line 104: “Proinflammatory Panel-1 (IFN-Ɣ, IL-1β, IL-2, IL-4, IL-6, IL-8, IL-10, IL12p70, IL-13, TNF-α)” – IL1Ra, IL-10 and IL-13 are anti-inflammatory markers.

“Proinflammatory Panel 1” is the name of the panel according to the manufacturers (MSD) and has been used here to make it easier for the reader to replicate the panel. 

Line 101-6: It would be better to write all the biomarker names out in full the first time they are used, since the abbreviations may not be familiar to all the readership of this journal.

The abbreviations used are standardised cytokine terminology and the authors would be concerned that writing them out in full might make the manuscript difficult to read. 

Line 110-1: “CMV viral loads were quantified by DNA PCR in a subset of participants with available plasma as described previously(20).” Please explain who this subset was or what a decision to test for CMV was based on.

This subset consisted of participants with stored plasma available for analysis. This sentence has been changed to provide clarity for the reader.

Write numbers in the beginning of a sentence out in full.

Corrected in the manuscript

Line 137: Were HIV-uninfected participants also screened for CMV infection?

Yes, participants without HIV also had CMV PCRs performed but did not have detectable virus.

Line 138: Was cfPWV only performed for the HIV-positive participants?

cfPWV was performed for all participants but the median for HIV positive patients is reported for context here. The effect of HIV on cfPWV was the subject of an earlier publication which has been referenced.

Line 151: “HIV cohort, who were also more likely to work 7 hours per week…” – should this be 7 days per week?

We thank the reviewer for highlighting this error which has now been corrected.

Table 1: Please check all numbers in Table 1. For instance, the categories for the floor variable do not add up to 100% for the HIV-infected participants. 

We thank the reviewer for point out this rounding error – this has been corrected.

In addition, 198/262 does not equal 80%; 71/262 does not equal 29%; 139/262 does not equal 56%; 90/105 does not equal 84%; 52/105 does not equal 51%; 64/105 does not equal 62%; etc, etc. Note that dark lines that divides the “mode of transport” category.

Whilst the total number of observations for the socio-economic questionnaires is given at the top of the table, the denominator for the percentages in the table may change for variables where some data are missing. This denominator can be derived from the percentage. 

Line 158-9: “cfPWV was higher for the 16 (8%) participants who travelled to work in a car compared to those who did not [8m/s (7.3 to 10.2)”. It is not clear what is being presented: median and interquartile range? This is also true later in the manuscript e.g. when monocyte data are presented.

This has been clarified in the manuscript.

Line 161: “p0.0005” should be p=0.0005.

Corrected in manuscript.

Line 162: “These factors remained predictive of cfPWV”. It seems that associated is the more appropriate word here since it does not appear as if predictive models had been constructed.

Corrected in manuscript. 

If “Carotid femoral pulse wave velocity (cfPWV) was also carried out at both time points as a

measurement of arterial stiffness”, which values are presented in lines 137 and 138, and which were used in the section “Socio-economic predictors of cfPWV amongst PLWH”? Had there been any change in cfPWV over time?

The analyses presented here are baseline cfPWV values, this has been corrected in the manuscript. Where changes over time were associated with socio-economic variables, the relationship has been reported in the relevant results section. For example, second paragraph of ‘Education level and income section’ – “Participants with a tertiary education tended to experience greater improvements in CD8 activation after 42 weeks of ART compared to those without any education [%change -5.8 (-16.7 to 2.2) versus 2.4 (-7.9 to 12); p=0.07], as did those on higher incomes [% change 6.0 (-8.2 to 20.1) versus -18.2 (-21.1 to -4.8) comparing lowest and highest income categories; p value=0.0001]. However, neither educational level nor patient income predicted change in the proportion of intermediate monocytes on ART (p=0.69 and 0.41 respectively).”

Since many readers will not be familiar with cfPWV, there should be some explanation about what higher values mean in terms of arterial stiffness.

An explanatory paragraph on this has been added to the introduction.

Line 174: What is meant by “an indirect correlation”? According to the statistics, no correlations had been performed.

This has been amended in the manuscript.

Line 181: “as did those on higher incomes” – should this be with higher incomes?

Corrected in manuscript

Lines 185-195: What is meant by the fold change values? Is this the change over time? Why was this selected rather than the median values for each time point? This is not explained in the statistical analysis section.

Values here are median change over time between baseline and exit visits, to provide an indication of how the proportion of T cells have changed on ART. This has been clarified in the text.

Line 191: Since IL-6 has been reported to be affected by obesity, was the association with higher income assessed for confounding with body mass index?

For this analysis, the relationship between inflammatory markers and socioeconomic status were adjusted for age and sex. The effect of obesity would be outside the scope of this paper, but would certainly be important to look at in future studies. 

Line 191-3: “Levels of IL6 and IL13 were also higher amongst people with higher incomes [2.9 μg/mL (1.13 to 7.6), p=0.028; and 3.1 μg/mL (1.2 to 8.3), p=0.025 respectively].” Why are the median values (I presume these are medians and IQRs?) shown here and not the fold change? Which time point is referred to? Why are the comparator values not shown for participants with lower incomes? Which groups were combined to create the group “higher incomes”?

These values are fold change with 95% CI, in keeping with the values reported in the previous sentence, this has been repeated for clarity. The time point is at baseline, as for all cross-sectional analysis throughout the manuscript. The comparator >25USD per month to <25 USD per month has been made clearer in the text. 

Line 193: What is meant by “Adjusted nonclassical and intermediate monocytes”?

This has been clarified in the manuscript. 

Line 211-4: “Baseline CD8 activation remained significantly higher amongst those who grew crops in adjusted analysis [fold change 12.1% (5.2 to 213 19.0);p=0.001], with significantly higher adjusted levels of IL7 and IL8 [IL7 fold change 214 2.47pg/mL (95%CI 1.10 to 5.52),p=0.028; IL8 2.97pg/mL (1.10 to 7.97),p=0.031].” Where are comparator values for participants who did not grow crops? The same is true for lines 231-5.

These values are fold change from regression analysis comparing values among those who did grow crops to values for those who did not. A separate comparator group is therefore not presented. 

A table with the descriptive results, as well as the unadjusted and adjusted results of the univariate and multivariable analysis of all the biomarkers tested would be very helpful and greatly improve understanding of the results. One would like to see the results of all the markers that had been tested. In addition, results should be displayed in a systematic fashion since now only results that the authors seemingly found interesting are displayed, even in the supplementary tables. This makes a full assessment of the results impossible.

Due to the volume of data it is challenging to present all of the univariate and multivariate analyses together, and therefore we preferred to present those results that were statistically significant, biologically important and relevant to the message of the paper. To help provide clarity we have provided a new table in supplementary data showing median values of plasma biomarkers for each socio-economic variable. 

Line 231: “After adjustment for confounders…”. It is not clear what confounding had been adjusted for with regards to CD8 activation.

CD8 activation is adjusted for age and sex, and is outlined in the methods statement as well as the supplementary material. 

Had any adjustment been made for CMV co-infection?

Adjustment was not made for CMV as planned multi-variate analysis did not include CMV as a confounder or mediator.

Monocytes were only defined as classical (CD14++CD16-), intermediate (CD14++CD16+) or nonclassical (CD14+CD16+) and no markers of activation had been included in the flow cytometric analysis. For this reason, any statement about monocyte activation is incorrect and should be rephrased and re-interpreted according to the phenotype observed. Just two examples where this is problematic are the following: “population did not show improvements in monocyte activation on ART.”; “Intermediate monocytes were expanded amongst four PLWH who used an unprotected well as their main water source. Although this group is very small, this finding is in keeping with low socio-economic status and unsafe water sources being associated with monocyte activation and innate inflammatory pathways(29).” As it stands, it is completely unclear what it means to have a higher proportion of intermediate monocytes, for instance.

On activation, monocytes express CD16, signally a proinflammatory phenotype. For clarity, the term monocyte activation has been replaced in the manuscript in the two instances that the term was used, as outlined by the reviewer. 

It is not clear what the purpose of the control group is, since only socio-economic variables were compared between them and the HIV-infected participants.

The purpose of the control group here was to place socio-economic status for people living with HIV in the context of socio-economic status in Malawi.

Figure 1: there is no indication of what the asterisks mean.

A figure legend has been added to clarify this.

Figure 2 is very difficult to understand and should be augmented with a footnote that describes what is depicted.

A figure legend has been added to explain this.

Figure 4 would be greatly strengthened by the inclusion of supporting references. It is not clear how viral antigen stimulation leads to lymph node fibrosis.

A figure legend has been added to figure 4 with references and a clear explanation that it is a hypothesis which requires further research.

Why were CD4 cell markers not also assessed? Why are the data on T cell exhaustion and senescence not also shown?

Neither CD4 T cell data nor exhaustion / senescence markers demonstrated an association with socio-economic variables and are therefore not reported in line with the methodological approach outlined. It should be noted that for this population with low nadir CD4 count, few CD4 T cell events were captured. 

 

Reviewer #2: 

• What is a water kiosk? The authors report that it provides clean water but I would like a little more information on what it is.

We thank the reviewer for highlighting this. The functioning of a water kiosk has been clarified in the discussion.

• In line 151, should 7 hours be 7 days per week

This has been corrected in the manuscript. We thank the reviewer for raising this inconsistency. 

• The authors could include a brief comment on what is known on the effect of repeated social stress on myelopoiesis (e.g. www.pnas.org/content/110/41/16574)

We thank the reviewer for this helpful suggestion. This has been added to the discussion.

• I think a paragraph on what is known on SES and NCDs in low-income countries would also benefit the paper – e.g. referencing this article amongst others https://www.thelancet.com/journals/langlo/article/PIIS2214-109X(17)30054-2/fulltext 

Again, we thank the reviewer for this helpful suggestion and have added this point to the discussion.

---

## [Decision Letter · Decision Letter 1]

21 Jul 2021

PONE-D-20-21907R1

Inflammatory pathways amongst people living with HIV in Malawi differ according to socioeconomic status

PLOS ONE

Dear Dr. Kelly,

Thank you for submitting your manuscript to PLOS ONE. After careful consideration, we feel that it has merit but does not fully meet PLOS ONE’s publication criteria as it currently stands. Therefore, we invite you to submit a revised version of the manuscript that addresses the points raised during the review process.

We look forward to receiving your revised manuscript.

Kind regards,

Olalekan Uthman, MD, MPH, PhD, FRSPH, FHEA

Academic Editor

PLOS ONE

Journal Requirements:

Reviewers' comments:

Reviewer's Responses to Questions

**Comments to the Author**

1. If the authors have adequately addressed your comments raised in a previous round of review and you feel that this manuscript is now acceptable for publication, you may indicate that here to bypass the “Comments to the Author” section, enter your conflict of interest statement in the “Confidential to Editor” section, and submit your "Accept" recommendation.

Reviewer #1: (No Response)

Reviewer #2: All comments have been addressed

2. Is the manuscript technically sound, and do the data support the conclusions?

Reviewer #1: Partly

Reviewer #2: Yes

3. Has the statistical analysis been performed appropriately and rigorously? 

Reviewer #1: Yes

Reviewer #2: Yes

4. Have the authors made all data underlying the findings in their manuscript fully available?

Reviewer #1: No

Reviewer #2: Yes

5. Is the manuscript presented in an intelligible fashion and written in standard English?

Reviewer #1: Yes

Reviewer #2: Yes

6. Review Comments to the Author

Reviewer #1: Abstract: Please correct the following sentence: “Water kiosk use showed a protective association protective against T cell activation…”

Introduction: Please include the abbreviation for carotid femoral pulse wave when the term is used for the first in line 61: “Carotid femoral pulse wave velocity is a gold standard measurement of arterial stiffness and has been validated against clinical outcomes in high-income settings…”. Do not introduce the term in line 95.

In the previous round of reviews, I have requested the authors to write all the biomarker names out in full the first time they are used, since the abbreviations may not be familiar to all the readership of this journal. The authors had responded that “The abbreviations used are standardised cytokine terminology and the authors would be concerned that writing them out in full might make the manuscript difficult to read.” I do not agree. Since the journal is not an Immunology journal and has a wide readership, the terms have to be explained. In fact, explaining the terms, as I have illustrated below, only adds 5 lines to the manuscript and provides the non-immunologist with better understanding.

CD38/HLADR, PD1 and CD57 expression, respectively. Monocytes were defined as classical (CD14++CD16- ), intermediate (CD14++CD16+ ) or nonclassical (CD14+CD16+). Stored plasma was tested for 22 cytokines: Proinflammatory Panel-1 (IFN-Ɣ, IL-1β, IL-2, IL-4, IL-6, IL-8, IL-10, IL12p70, IL-13, TNF-α), Vascular Injury Panel-2 (SAA, CRP, VCAM-1, ICAM-1), Chemokine Panel-1 (MIP-1β, IP-10, MCP-1), Angiogenesis -Panel-1 (VEGF-A, bFGF) and single analyte assays for IL1R antagonist and IL-7.

VERSUS

Cluster of differentiation (CD)38/Human Leukocyte Antigen – DR isotype (HLADR), programmed death (PD)-1 and CD57 expression, respectively. Monocytes were defined as classical (CD14++CD16- ), intermediate (CD14++CD16+ ) or nonclassical (CD14+CD16+). Stored plasma was tested for 22 cytokines: Proinflammatory Panel-1 (interferon [IFN]-Ɣ, interleukin [IL]-1β, IL-2, IL-4, IL-6, IL-8, IL-10, IL12p70, IL-13, tumour necrosis factor [TNF]-α), Vascular Injury Panel-2 (serum amyloid A [SAA], C-reactive protein [CRP], vascular cell adhesion molecule [VCAM]-1, intercellular adhesion molecule [ICAM-1]), Chemokine Panel-1 (macrophage inflammatory protein [MIP]-1β, interferon γ-induced protein [IP]-10, monocyte chemoattractant protein [MCP]-1), Angiogenesis -Panel-1 (

vascular endothelial growth factor [VEGF]-A, basic fibroblast growth factor [bFGF]) and single analyte assays for IL-1receptor antagonist (IL-1Ra) and IL-7.

As requested previously, please write numbers out in the beginning of sentences, e.g. “15 (4%) participants died during the study”; “201 (55%) had electricity and 95 (26%) had a private water tap at home. 75 (21%) were unemployed and of those employed, 108 (29%) earned less than 25 USD per month, although 97 (36%) of participants were not able to provide an estimate of monthly salary. 339 (92%) of participants felt that their household did not have enough food.”

One of my previous comments was about Table 1, where the percentages were not correct according to the denominator provided. The authors responded as follows: “Whilst the total number of observations for the socio-economic questionnaires is given at the top of the table, the denominator for the percentages in the table may change for variables where some data are missing. This denominator can be derived from the percentage.” It is good scientific practice to indicate where data are missing and the authors should please make these amendments to the table.

I still have difficulty in understanding this section: “On univariate analysis of the HIV positive cohort, cfPWV was higher for the 16 (8%) participants who travelled to work in a car compared to those who did not [median (IQR) 8m/s (7.3 to 10.2) versus 7.2m/s (6.3 to 8.1); p=0.008]; for those who owned a television [7.7 (6.7 to 8.5) versus 7.2 (6.2 to 7.8) p=0.0005]; and for those who had an electricity supply at home [(7.4 (6.5 to 8.3) versus 7.2 (6.4 to 8.1) p=0.093]. These factors remained associated with cfPWV when adjusted for age, sex, blood pressure and haemoglobin: car ownership [fold change 1.3m/s (95% CI 1.10 to 1.56); p=0.003]; television ownership [1.12m/s (1.03 to 14 174 1.23); p=0.012]; and electricity access [1.09m/s (1.01 to 1.17); p=0.029].” For the univariate analysis, it seems that the baseline cfPWV is used, but for the multivariable analysis, which I assume are the results given after adjusting for age, sex, blood pressure and haemoglobin, the fold change of the cfPWV is given. What is the rationale for that?

It is unclear how the authors decided whether a change in CD8 activation is an improvement or a deterioration: “Participants with a tertiary education tended to experience greater improvements in CD8 activation after 42 weeks of ART compared to those without any education [%median (IQR) change from baseline to 42 weeks -5.8 (-16.7 to 2.2) versus 2.4 (-7.9 to 12); p=0.07], as did those with higher incomes [% change 6.0 (-8.2 to 20.1) versus -18.2 (-21.1 to -4.8) comparing lowest and highest income categories; p value=0.0001].” This is especially true since no control values are shown and there are no reference values of CD8 activation one can refer to. So, while one group can have higher or lower levels, how does one know which one is better?

Similarly, how did the researchers decide what is an improvement in monocyte phenotype: “Compared to those with a brick floor, participants with an earth floor had higher proportions of nonclassical monocytes at presentation [mean (IQR) 16.9% (9.9 to 25.80) versus 10.3% (8.2 – 16.7); p=0.04] and were also more likely to experience improvements after 42 weeks of ART [nonclassical monocyte mean (IQR) change -2.2% (-12.0 to 2.9) versus 5.8% (3.9 to 7.2); p=0.026].”

“Adjusting for age and sex, we first compared those with some education to those without any: IL7 was lower for those with some education [fold change 0.20µg/mL (0.07 to 0.59); p=0.002] who also trended towards lower bFGF [fold change 0.22µg/mL (0.048 to 1.09); p=0.056] and higher proportion of activated CD8 T cells [fold change 7.02% (-0.97 to 15.0); p=0.085].” I am very sorry if I am just not understanding this. The authors state that IL-7 was lower, but show the fold change in support. Do they therefore mean that the fold change was greater in this group or do they mean that the exit value was lower? Even after re-reading the statistics section several times, I still fail to understand what the authors are trying to show.

“Interestingly, an indirect association was observed with intermediate monocytes, with higher median (IQR) amongst those with no education compared to a tertiary education [12.6% (5.4 – 15.5) versus 7.3% (5.3 – 9.8); p=0.01] …” Do the authors mean an inverse association instead of an indirect association?

A limitations that has to be mentioned is that there was no adjustment for other clinically relevant parameters such as obesity. This is especially important since some of the findings, such as “Levels of IL6 and IL13 were also higher amongst people with higher incomes [fold change (95% CI) 2.9 µg/mL (1.13 to 7.6), p=0.028; and 3.1 µg/mL (1.2 205 to 8.3), p=0.025 respectively]” could be severely confounded by body weight.

Since non-classical monocytes are associated with a less inflammatory phenotype and also with tissue repair, how do the authors explain the following finding? “Conversely CD16 positive monocytes, both non-classical and intermediate, were expanded amongst those with variables consistent with a lower socio-economic status including lower income, less education and earth flooring. This expansion of CD16 positive monocytes did not resolve on ART.” This is also contradicted in the following statement: “Participants in lower socio-economic groups experience expanded inflammatory monocytes, which did not improve on ART.” While monocyte phenotypes are extraordinarily complex and still a topic of debate, it is generally accepted that classical monocytes (CD14+ CD16-) are the most pro-inflammatory. See for instance Kapellos TS et al. Human Monocyte Subsets and Phenotypes in Major Chronic Inflammatory Diseases. Front. Immunol., 30 August 2019 | https://doi.org/10.3389/fimmu.2019.02035

It is not clear why CMV was tested for but not adjusted for in the multivariable analysis since CMV is known to significantly affect CD8+ T-cell and monocyte populations, and especially since there was an association in this study with water use, suggesting that some of the findings related to water use could be confounded by CMV. To list but two relevant references in this regard:

van den Berg SPH, Pardieck IN, Lanfermeijer J, et al. The hallmarks of CMV-specific CD8 T-cell differentiation. Med Microbiol Immunol. 2019;208(3-4):365-373. doi:10.1007/s00430-019-00608-7

Baasch S, Ruzsics Z, Henneke P. Cytomegaloviruses and Macrophages—Friends and Foes From Early on? Front. Immunol., 12 May 2020 | https://doi.org/10.3389/fimmu.2020.00793

At the very least, this should be stated as a limitation.

Can the authors hypothesize as to how the following markers can be reconciled, especially since higher levels of IL-12p70 are known to enhance the cytotoxic effects of CD8+ T-cells?

“After adjustment for confounders, CD8 activation remained lower amongst water kiosk users [fold change -7.05% (95%CI -12.1 to -1.97); p=0.007]. MIP1β, sICAM1 and sVCAM1 were also all significantly lower amongst kiosk users [fold change 0.63pg/mL (95%CI 0.40 to 254 0.99), p=0.047; 0.65µg/mL (0.44 to 0.59), p=0.026; and 0.74 µg/mL (0.57 to 0.97), p=0.03 respectively] but IL12p70 was higher [2.39 (1.34 to 4.28); 0=0.003].” Please also correct 0=0.003.

Reviewer #2: All comments have been addressed, and the ms is ready for publication. Congratulations to the authors.

7. PLOS authors have the option to publish the peer review history of their article (what does this mean?). If published, this will include your full peer review and any attached files.

Reviewer #1: No

Reviewer #2: No

---

## [Author Response · Author response to Decision Letter 1]

22 Jul 2021

Please see Response to Reviewers letter at the end of the PDF for review.

---

## [Editor Report · Decision Letter 2]

11 Aug 2021

Inflammatory pathways amongst people living with HIV in Malawi differ according to socioeconomic status

PONE-D-20-21907R2

Dear Dr. Kelly,

We’re pleased to inform you that your manuscript has been judged scientifically suitable for publication and will be formally accepted for publication once it meets all outstanding technical requirements.

Kind regards,

Olalekan Uthman, MD, MPH, PhD, FRSPH, FHEA

Academic Editor

PLOS ONE

---

## [Editor Report · Acceptance letter]

16 Aug 2021

PONE-D-20-21907R2 

Inflammatory pathways amongst people living with HIV in Malawi differ according to socioeconomic status 

Dear Dr. Kelly:

I'm pleased to inform you that your manuscript has been deemed suitable for publication in PLOS ONE. Congratulations! Your manuscript is now with our production department. 

Kind regards, 

on behalf of

Dr. Olalekan Uthman 

Academic Editor

PLOS ONE